

# Beyond the prey: male spiders highly invest in silk when producing worthless gifts

Camila Pavón-Peláez[1,2], Valentina Franco-Trecu[2], Irene Pandulli-Alonso[3], Therésa M. Jones[4] and Maria J. Albo[3,5]

[1] Programa de Desarrollo de las Ciencias Básicas (PEDECIBA), Universidad de la República, Uruguay
[2] Departamento de Ecología y Evolución, Facultad de Ciencias, Universidad de la República, Montevideo, Uruguay
[3] Departamento de Ecología y Biología Evolutiva, Intstituto de investigaciones Biológicas Clemente Estable, Montevideo, Uruguay
[4] School of BioSciences, The University of Melbourne, 3010 Victoria, Australia
[5] Sección Entomología, Facultad de Ciencias, Universidad de la República, Uruguay

## ABSTRACT

In the spider *Paratrechalea ornata*, males have two gift-giving mating tactics, offering either a nutritive (prey) or a worthless (prey leftovers) silk wrapped gift to females. Both gift types confer similar mating success and duration and afford males a higher success rate than when they offer no gift. If this lack of difference in the reproductive benefits is true, we would expect all males to offer a gift but some males to offer a worthless gift even if prey are available. To test this, we allowed 18 males to court multiple females over five consecutive trials. In each trial, a male was able to produce a nutritive gift (a live housefly) or a worthless gift (mealworm exuviae). We found that, in line with our predictions, 20% of the males produced worthless gifts even when they had the opportunity to produce a nutritive one. However, rather than worthless gifts being a cheap tactic, they were related to a higher investment in silk wrapping. This latter result was replicated for worthless gifts produced in both the presence and absence of a live prey item. We propose that variation in gift-giving tactics likely evolved initially as a conditional strategy related to prey availability and male condition in *P. ornata*. Selection may then have favoured silk wrapping as a trait involved in female attraction, leading worthless gift-giving to invade.

# INTRODUCTION

In some invertebrate species, individuals (usually males) offer nuptial gifts to potential mates (usually females) increasing their reproductive success (*Vahed, 1998*; *Lewis et al., 2014*). The diversity of gifts includes endogenous materials, such as male-produced glandular secretions or proteins in their ejaculates, to exogenously obtained gifts such as nutrient-rich prey or even inedible items (*Vahed, 1998*; *Lewis et al., 2014*). Studies of gift-giving species typically demonstrate that males offering gifts gain greater access to matings and have higher paternity compared to males that do not offer a gift, while females may gain nutritive benefits (*Vahed, 1998*; *Lewis et al., 2014*). The ability to provide a nuptial gift

Corresponding author
Maria J. Albo, mjalbo-grana@gmail.com

may be dependent on the condition of the giver thus acting as an honest indicator of some aspect of individual quality (*e.g.*, good hunter, provider) (*Dougherty, 2021*). In fact, limited (or no) gift production is often associated with a reduction in a male's nutritive status and current physiological condition, ultimately resulting in a decrease of their reproductive success and lifespan (*Downes, 1970*; *Thornhill, 1976*; *Jia, Jiang & Sakaluk, 2000*; *Wedell & Ritchie, 2004*; *Engels & Sauer, 2006*; *Hall, Bussière & Brooks, 2009*; *Albo, Toft & Bilde, 2011*; *Albo et al., 2014*; *Macedo-Rego et al., 2016*). A general conclusion from these studies is that the probability of nuptial gift production likely varies with food availability resulting in plastic, environmentally-determined behavioural responses (*Pigliucci, 2001*; *Charmantier et al., 2008*; *Cornwallis & Uller, 2010*).

Gift-giving plasticity creates a scenario for the origin and maintenance of alternative mating tactics (AMTs), which are discrete individual responses that lead to tactic-specific differential reproductive success (*Gross, 1996*; *Brockmann, 2001*; *Taborsky, Oliveira & Brockmann, 2008*; *Engqvist & Taborsky, 2016*). The vast majority of AMTs are a conditional strategy in which the expression of the tactics, either dominant or alternative, has a strong environmental influence. The alternative tactic is less successful than the dominant and therefore, typically occurs at a lower frequency within the population (*Gross, 1996*; *Brockmann, 2001*; *Taborsky, Oliveira & Brockmann, 2008*; *Engqvist & Taborsky, 2016*). For example, in empidid flies, courting males join together in a swarm each carrying a captured insect prey (nuptial gifts) that function as a stimulus for the females to choose the mate (*Kessel, 1955*). Empidid males without gifts are usually rejected by females; but males that offer inedible items accrue some mating success albeit less than males offering nutritive prey gifts (*Preston-Mafham, 1999*). This has been also shown in the gift-giving spider, *Pisaura mirabilis*. In this species, males provide females either a genuinely nutritive gift (*i.e.,* fresh prey –the dominant tactic) or a worthless gift (*i.e.,* insect exoskeletons –alternative tactic). Worthless gifts are given approximately 30% of the time and males incur paternity costs as a result of a reduction in mating duration (*Albo et al., 2011*; *Albo et al., 2019*). These costs rise because mating duration aligns with the time taken for a female to completely consume the gift contents (approximately 1 h, *Albo et al., 2011*). Male *P. mirabilis* investment into gift production is dependent on its type: for nutritive gifts (*i.e.,* dominant tactic), male effort is realised through prey capture; for worthless gifts (*i.e.,* alternative tactic) males switch effort towards a higher investment in silk wrapping (*Ghislandi et al., 2017*). As in the case for empidids, female *P. mirabilis* penalize males that offer worthless gifts indicating that gift content is relatively more important than the silk (*Albo et al., 2011*).

For some species, the act of giving a gift seems to be as important as the content of the gift. For example, in the spider *Paratrechalea ornata* worthless gifts are frequently (70% of males) offered to females (*Albo et al., 2014*), suggesting that this gift type can be a dominant rather than an alternative tactic. For such an evolutionary outcome, several changes would have been necessary to occur. First, it is likely that worthless gifts initially appeared as an alternative tactic allowing few males to increase their mating chances when prey were limited (*Albo et al., 2014*). Early studies confirm that individual males are plastic and adjust their mating tactic according to prey availability and their own body condition (*Albo et al., 2014*; *Trillo, Melo-González & Albo, 2014*). Additionally, when encountering females

(especially mated females) in the absence of prey, some males produce a worthless gift, rather than no gift, which substantially increases their mating success (*Albo & Costa, 2010*; *Albo et al., 2014*). Second, for this mating tactic to successfully spread in high numbers in a population, it must have provided males with similar reproductive success as nutritive gifts. A possibility is that, through the silk wrapping males reinforce the gift attraction as an honest signal of their quality, leading females to switch their preferences for the food gift (*i.e.,* the content is not further relevant) or become under exploitation (*Arnqvist & Rowe, 2005*; *Vahed, 2007*). In accordance with this, silk wrapping does play a relevant role in female attraction (*Brum, Costa-Schmidt & De Araújo, 2012*; *Trillo, Melo-González & Albo, 2014*). There is currently no evidence that females discriminate between worthless or nutritive gifts as they are as equally likely to mate (and for comparable durations) with males offering either gift type (*Albo et al., 2014*; *Pandulli-Alonso, Quaglia & Albo, 2017*). Further, mating duration is extremely short in this species giving no time for spider females to complete the gift consumption during sperm transfer (*Albo et al., 2014*).

Based on the assumption that when a *P. ornata* male offers a worthless gift he gains similar reproductive benefits to when he offers a nutritive gift, we would expect individuals choosing to offer worthless gifts even when they have the opportunity to produce a nutritive gift. This has never been explicitly tested. Further, if silk wrapping of the gift is reinforcing the signal acting as a reliable quality indicator of the potential mate, we predict that males would invest more in silk wrapping when offering worthless gifts than when the gift is nutritive; and that this result would hold for worthless gifts produced in the presence or absence of prey. Here, we examined within-individual male mating tactics in *P. ornata* by repeatedly exposing males to sexually mature females in the presence of live prey (nutrient gift) and insect exoskeletons (worthless gift), thus providing males the opportunity to produce either gift type on multiple occasions. We expected that males would be equally likely to produce either gift type. If silk investment is inexpensive compared to the prey capture activity, then males should produce worthless gifts even in the presence of a prey. Such a result would indirectly demonstrate that males have no restrictions on their reproductive benefits in relation to the gift type offered to females. Silk investment in worthless gifts is also predicted to be similar when males are in the presence or absence of prey. Thus, we compared the results from trials with nutrient-rich prey with trials in which we exposed males to females in the absence of prey but with insect exoskeletons. But, as silk wrapping is condition dependent (*Trillo, Melo-González & Albo, 2014*), a male's decision at any gift giving opportunity may be triggered by their size or body condition, resulting in consistent individual responses. In contrast, if the gift content is more relevant for males in terms of reproductive success (*i.e.,* females exert cryptic female choice that favors nutritive gifts) and/or silk investment is expensive, they should try to capitalize their effort in prey capture to produce nutritive gifts as often as possible.

## MATERIAL AND METHODS

*Paratrechalea ornata* is a semi-aquatic spider found along the edge of streams and rivers from southern Brazil to southern Uruguay. Individuals are crepuscular/nocturnal and
can be seen on stones and pebbles emerging from the water (*Costa-Schmidt, Carico & De Araújo, 2008*). The typical male courtship behaviour is triggered when detecting female silk (and associated pheromones) on the substrate (*Albo, Costa-Schmidt & Costa, 2009*). At that time, the male vibrates his forelegs and pedipalps while walking along searching for an item. This item may be flying insects emerging from the watercourses (*i.e.,* Ephemeroptera), which are detected mostly through wing vibrations, or an inedible item like prey leftovers or plant parts that the male has eaten or has actively searched among the pebbles, respectively. After securing an item, the male usually wraps it in silk prior to offering it to a female (*Albo & Costa, 2010*; *Albo et al., 2014*). Silk wrapping of the item includes several wrapping bouts, and in each one the male produces a silk sheet in the substrate, deposits the item on it and afterwards covers it with silk (*Costa-Schmidt, Carico & De Araújo, 2008*; *Albo, Costa-Schmidt & Costa, 2009*). Once the male finishes the silk wrapping he walks (vibrating forelegs) with the gift in his chelicerae searching for a female. When offering the nuptial gift, the male adopts a particular position called the "hyperflexion" in which he presents the nuptial gift to the female (*Costa-Schmidt, Carico & De Araújo, 2008*). If a female is receptive, she grabs the gift and the male initiates pedipalp insertions (sperm transfer) while holding the gift with the claws of his third pair of legs. Between each pedipalp insertion the male returns to the face-to-face position and holds the gift again prior to continuing. Once mating and sperm transfer are complete, the pair move apart and the female leaves with the gift (*Costa-Schmidt, Carico & De Araújo, 2008*; *Albo & Costa, 2010*).

## Spider collection and maintenance

We collected juveniles and subadults ($N = 78$) between 21:00 and 23:00 h during the reproductive season of 2015 (August-September). Individuals were sampled from a well-studied population from Minas, Uruguay (34°16′40.10′S, 55°14′00.80′W), in which the percentage of worthless gifts ranges between 30–80% in the field (MJ Albo unpublished data). To accelerate their development, spiders were initially transferred to individual plastic jars (eight cm diameter and seven cm height) and held in a temperature-controlled room at 24.5 °C ($\pm 0.52$ SE). We fed individuals with five fruit flies (*Drosophila* sp.) three times a week and thus, individuals were maintained under similar conditions as in the field, with moderate prey available before the trials. We provided *ad libitum* water daily through a moistened cotton wool ball. Once individuals reached adulthood, we transferred them to the experimental room (average $\pm$ SE: 20.3 $\pm$ 0.18 °C). We continued the same feeding regimen for adult spiders for the duration of the experimental period. To ensure all spiders were sexually mature and the females receptive (*Klein, Trillo & Albo, 2012*), unmated adult males and females commenced trials between four and five weeks after their final moult (females: 30 days $\pm$ 2.76 SD; males: 33 days $\pm$ 2.20 SD). Following the completion of all trials, we measured the males size (cephalothorax width) under a stereomicroscope (*Jakob, Marshall & Uetz, 1996*).

## Experimental design
### Likelihood of mating tactics: nutritive and worthless gifts

To examine whether male mating tactics (nutritive/worthless) are equally likely to co-occur, we repeatedly allowed males ($N = 36$) to court different novel females once every four days

over five consecutive trials (Nutritive-Worthless treatment group). We used 42 females that were randomly exposed several times (range 2–8) to different males. This represents the natural environment as females in this species readily mate multiply (*Pandulli-Alonso, Quaglia & Albo, 2017*). We simulated a typical courting condition observed in the field (MJ Albo personal observations) by exposing males to inedible items along the substrate and a flying prey. Thus, in the Nutritive-Worthless group ($N = 18$ males), males were provided with a live housefly (*Musca domestica)* and exuviae of a mealworm (*Tenebrio molitor* larva), which allowed them to offer either a silk-wrapped nutrient rich gift or a worthless gift.

We performed the trials in transparent glass cages (30 ×14 cm base, 20 cm height) in which we simulated natural conditions by covering the bottom with pebbles and water. Following previous protocols (*Albo et al., 2014*; *Albo & Peretti, 2015*), we placed a female in the experimental cage 24 h before introduction of a male, allowing her to habituate and deposit silk that stimulates male courtship and gift production (*Albo, Costa-Schmidt & Costa, 2009*). At the start of a trial, a male was placed with the female in the cage along with six mealworm exuviae distributed along the bottom of the cage. To create the scenario where a male could produce either a nutritive or a worthless gift in a single trial, we allowed each male access to both a prey item and an exuviae simultaneously. Once the males started courting females (defined by fast vibrations of forelegs), a live housefly was closely approached to the male *via* tweezers. This procedure has proved effective for this species as the male can detect the wing vibrations and capture the fly (*Albo, Costa-Schmidt & Costa, 2009*; *Albo & Costa, 2010*; *Brum, Costa-Schmidt & De Araújo, 2012*; *Albo et al., 2014*; *Albo & Peretti, 2015*); additionally allowing us to both control that all males had access to the prey at the same time and meant that the female was not able to capture it herself. All males were presented with a housefly as a first choice to produce a gift. We repeated the action of hand-giving the housefly every 15 min until the male grabbed and captured the fly, or grabbed an exuviae from the pebbles. We finished the trial when the male wrapped an item in silk and offered it to the female. If the male wrapped an item in silk but did not offer it to the female, we finished the trial an hour after the male ended the silk wrapping (*i.e.*, the last silk wrapping bout). The cases where males grabbed an item but did not wrap it in silk or failed to grab any item were not included for data analyses. In all cases, to ensure spiders remained unmated throughout their trials, we prevented males from mating using a paintbrush. During each trial, we recorded which gift item (*i.e.,* prey or exuviae) males utilized during courtship.

### Silk investment in worthless and nutritive gifts

Using the Nutritive-Worthless group, we measured the duration of silk wrapping, as well as the number of silk wrapping bouts. We calculated total silk wrapping duration (min) as the sum of all wrapping bouts durations. To compare a male's silk investment in the presence and absence of prey, we conducted one further treatment (Worthless group) and examined the results against those from the Nutritive-Worthless group. In the Worthless group, we exposed 18 males to five consecutive females in the absence of a nutrient rich prey but in the presence of exuviae. We obtained data from 64 out of 90 trials in which males produced a worthless gift. Aside from offering a living fly with tweezers, we followed

the same experimental protocol described above, so males in both treatment groups had comparable courting opportunities but differed in the type of gift they could produce.

## Data analysis

We used Generalized Linear Mixed Models (GLMM) (*Pinheiro & Bates, 2006*) in R free software (*R Core Team, 2019*). Among the running models with the same random structure, we assessed each simplification with the Akaike Information Criterion (AIC) using the $\Delta$AIC >2 criterion (*Bolker, 2007*). All selected models were subject to the customary residual analyses (*Pinheiro & Bates, 2006*) and were found to have a satisfactory fit. All initial models included female ID as random effect, but this variable was discarded due to non-significant results.

### Likelihood of mating tactics: nutritive and worthless gifts

We analysed data of the Nutritive-Worthless group, using a GLMM with binomial distribution (logit link function) to test the probability of producing a worthless gift (wrapped an exuviae in silk) against the data of nutritive gifts, including male size and the trial number (male prior behaviour) as fixed effects. The model included male identity as a random effect and intercept.

Considering the repeated measurements of males, we also checked for the repeatability in nuptial gift production (item used), duration of silk wrapping and number of silk-wrapping bouts in the Nutritive-Worthless group in which males can produce the two gift types. We estimated the individual repeatability for all reported response variables using LME with REML implemented with the rptR package (*Nakagawa & Schielzeth, 2010*). We included gift type as a fixed effect when analysing variables related to silk investment (duration of silk wrapping and number of silk-wrapping bouts). This analysis describes the relative partitioning of variance into within and between treatment group sources of variance, and the measure obtained (repeatability - *R)* ranges from 0 (high within-individual variance or low-repeatability) to 1 (low within-individual variance or high-repeatability). The repeatability in behavioural variables was classified into three categories: poorly repeatable ($R < 0.25$), moderately repeatable ($0.25 < R < 0.50$) and highly repeatable ($R > 0.50$). In this method, uncertainty is quantified *via* parametric bootstrapping (1000 times).

### Silk investment in worthless and nutritive gifts

We assessed variation in silk investment using GLMM with Gamma and Poisson distributions (log-link function), by examining duration of silk wrapping (min) and number of silk wrapping bouts, respectively. First, we examined the variation in silk investment in both gift types (nutritive/worthless) using the Nutritive-Worthless group. This model included gift type, male size and trial number as fixed effect and male identity as random effect. Second, we analysed the silk investment in worthless gifts under the presence or absence of prey by comparing the data from the Nutritive-Worthless and Worthless groups. This model included treatment group, male size and trial number as fixed effects and male identity as random effect.

**Table 1 Likelihood of producing a worthless gift.** Parameter estimates (SE) and *p*-values using Generalized Linear Mixed Models with a Binomial distribution (logit link function) to test the probability of producing a worthless gift (wrapped an exuviae in silk) according to the male size and the trial number in the Nutritive-Worthlesss group. The completed model is shown due to no significant effect was found. The model includes male identity (MaleID) as random effect (included as intercept).

| Fixed effects | | | |
|---|---|---|---|
| | **Intercept** | **Male size** | **Trial number** |
| **Worthless gift production** | Estimate = −10.13 (6.37) $p = 0.11$ | Estimate = 2.12 (1.45) $p = 0.15$ | Estimate = −0.10 (0.22) $p = 0.66$ |
| **Random effects (Male ID)** | | | |
| **Worthless gift production** | Variance = 1.19 SD = 1.09 | | |

## RESULTS

### Likelihood of mating tactics: nutritive and worthless gifts

In trials where males were allowed access to both a prey item and exuviae (Nutritive-Worthless) males produced a nutritive gift in 53 opportunities (mean number of trials per male $\pm$ SE = 2.94 $\pm$ 0.39) and in 18 trials (20%) they produced a worthless gift (mean number of trials per male $\pm$ SE = 1.00 $\pm$ 0.22). The likelihood of producing a worthless gift was not related either to the male size or the trial number (Table 1).

Individual male responses across the five consecutive trials varied (see Fig. 1). The type of item used by males for nuptial gift production revealed that when males were offered a choice between a nutritive and a worthless gift, 39% of males consistently wrapped houseflies in silk during all five trials; 50% wrapped either houseflies or exuviae across their trials; 11% always wrapped exuviae only. Given that half of the males varied their responses, the individual repeatability in gift type production (using nutritive or worthless gifts) for this treatment group was low (Table 2). Gift investment in silk, either the wrapping duration or number of wrapping bouts, differed within males resulting in low individual repeatability (Table 2).

### Silk investment in worthless and nutritive gifts

Comparing silk investment between both gift types by using the Nutritive-Worthless group, we found that males invested more time in silk wrapping (Fig. 2A) and engaged in more silk wrapping bouts (Fig. 2B) (Table 3) when producing worthless compared to nutritive gifts. Silk investment in worthless gifts did not differ between males in the presence (Nutritive-Worthless group) or in the absence of prey (Worthless group) (Table 4).

We did not find an effect of either male size or trial number on any of these parameters measured for the Nutritive-Worthless group (Table 3). But when comparing worthless gifts production from both groups, we found that silk wrapping duration was positively related to male size (Fig. 3, Table 4).
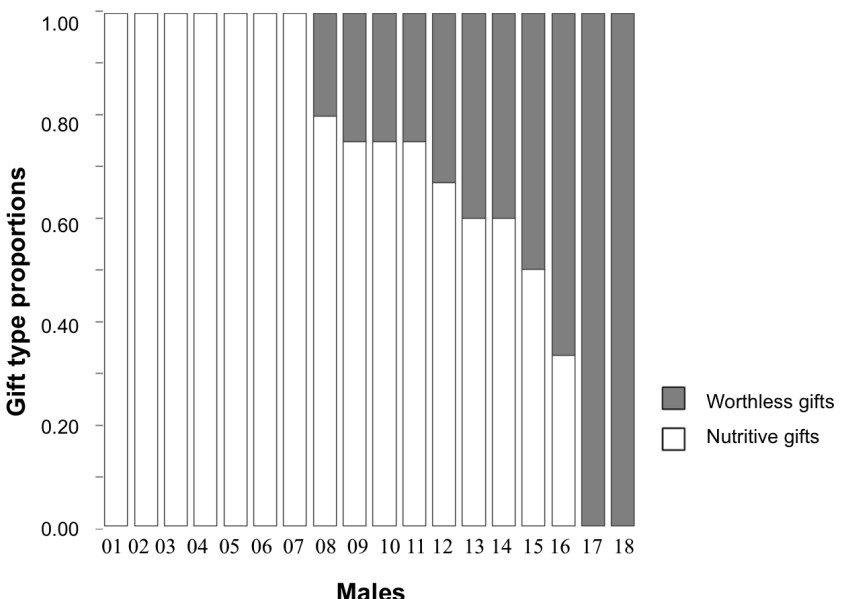

**Figure 1** **Proportions of gift type produced by each individual male of *Paratrechalea ornata*.** For each individual, the proportions of nutritive gifts are shown in white and worthless gifts in grey. These data are from the Nutritive-Worthless group ($n = 18$) in which males had the opportunity of producing a nutritive or a worthless gift. The bars show that 39% of the males always produced a nutritive gift, 11% always produced a worthless gift and 50% of the males showed a variable response.

**Table 2** **Individual behavioural repeatability.** Estimates of individual repeatability for the variables measured in the Nutritive-Worthless group. Repeatability (R) ranges from 0 (high within-individual variance or low-repeatability) to 1 (low within-individual variance or high-repeatability). The confidence interval (CI) was estimated at 95%.

| | Nutritive-Worthless group | | |
|---|---|---|---|
| | R | CI | p |
| **Gift type production** | 0.14 | 0–0.33 | 0.08 |
| **Silk wrapping duration** | 0.05 | 0–0.18 | 0.24 |
| **Number of silk wrapping bouts** | 0.08 | 0–0.21 | 0.16 |

## DISCUSSION

Here, we demonstrated that, even when *P. ornata* males had the opportunity to produce a nutritive gift, 20% of them chose to offer worthless gifts. However, the data indicate that males are plastic in their mating tactics: there was a low degree of repeatability and substantial switching between producing nutritive and worthless gifts or vice versa over consecutive trials. Further, we showed that, when producing worthless gifts, males invest more in silk compared to nutritive gifts. This finding agrees with our expectation that males are likely to produce either gift type investing in nutritive content or extra-investment in silk and agrees with early studies showing that they equally secure matings (*Albo, Toft & Bilde, 2014*; *Pandulli-Alonso, Quaglia & Albo, 2017*). Worthless gifts have probably evolved
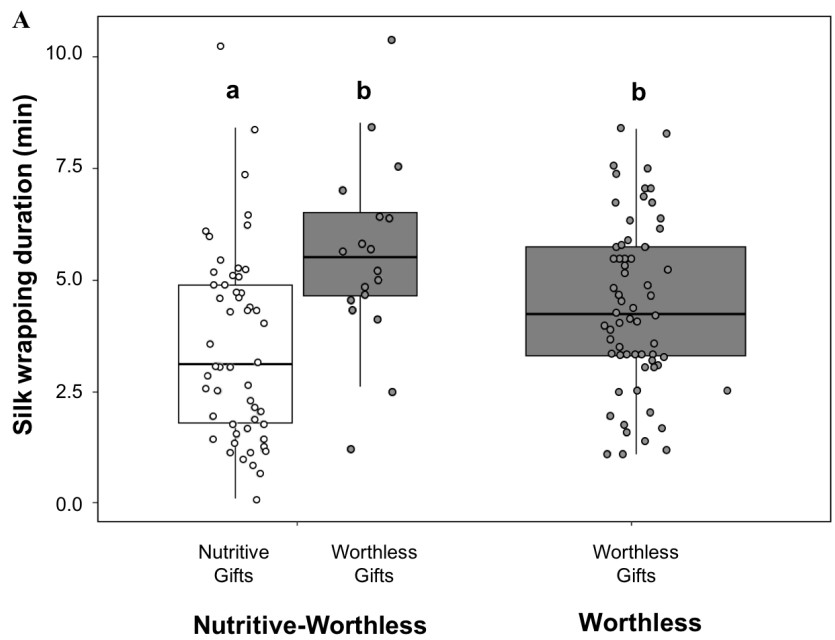

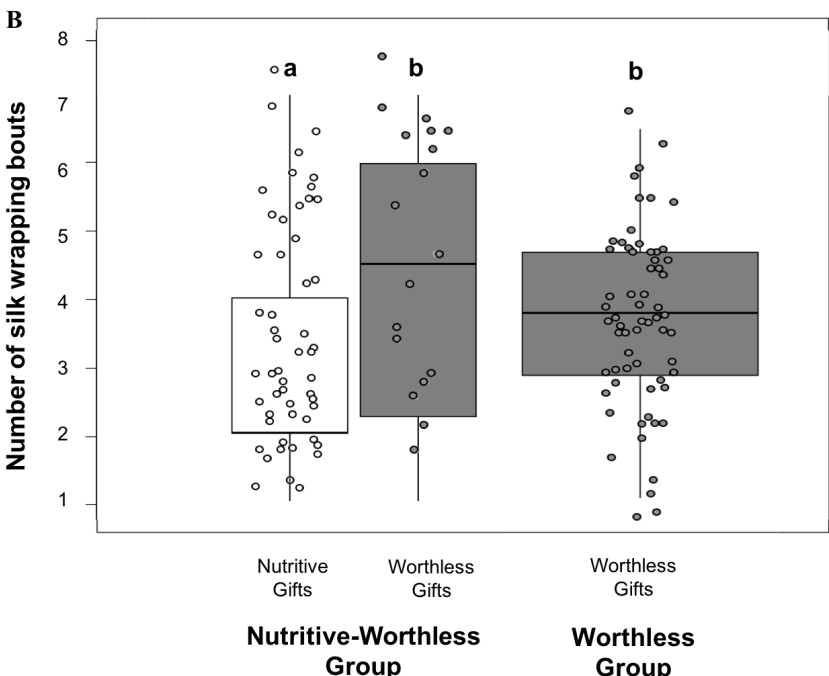

**Figure 2** **Males silk investment in nuptial gifts.** Differences in silk investment by *Paratrechalea ornata* males in relation to the experimental groups (Nutritive-Worthless/Worthless) and the gift type (Nutritive/Worthless gift). (A) Differences in silk wrapping duration (min) and (B) differences in number of silk wrapping bouts. Data of nutritive gifts are shown as white points and worthless gifts as grey points. First and third quartile are represented with boxes, the median is shown as thick horizontal line and the outliers as black points. Males from the Nutritive-Worthless group ($n = 18$) had the opportunity to produce nutritive or worthless gifts, and males from the Worthless group ($n = 18$) had the opportunity to produce only worthless gifts. Different letters indicate significant differences ($p < 0.05$).

**Table 3  Silk investment in nutritive and worthless gifts.** Parameter estimates and *p*-values (with significance shown in bold) using Generalized Linear Mixed Models with a Gamma or Poisson distribution and a log link function to explain differences in the duration of silk wrapping and number of silk wrapping bouts according to the gift type (nutritive/worthless) used by males in the Nutritive-Worthless group. The initial model included also male size, trial number and the gift type*male size interaction. Model selection was performed by the Akaike Information Criterion (AIC).

| Silk wrapping duration | | Intercept | Gift type (Worthless Gift) | Male size | Trial number | Gift type: male size | ΔAIC |
|---|---|---|---|---|---|---|---|
| **Model 1** | Estimate (SE) | 2.26 (1.46) | −2.51 (2.44) | −0.24 (0.34) | −0.002 (0.05) | 0.68 (0.56) | 5 |
| | *p* | 0.12 | 0.30 | 0.48 | 0.97 | 0.22 | |
| **Model 2** | Estimate (SE) | 1.50 (1.32) | 0.48 (0.18) | −0.06 (0.31) | 0.006 (0.05) | – | 4 |
| | *p* | 0.26 | **0.007** | 0.83 | 0.90 | – | |
| **Model 3** | Estimate (SE) | 1.51 (1.32) | 0.48 (0.18) | −0.06 (0.31) | – | – | 2 |
| | *p* | 0.25 | **0.007** | 0.84 | – | – | |
| **Model 4** | Estimate (SE) | 1.24 (0.10) | 0.47 (0.17) | – | – | – | 0 |
| | *p* | <0.0001 | **0.006** | – | – | – | |
| **Number of silk wrapping bouts** | | Intercept | Gift type (Worthless Gift) | Male size | Trial number | Gift type: male size | ΔAIC |
| **Model 1** | Estimate (SE) | 1.02 (0.17) | 0.28 (0.16) | −0.04 (0.09) | 0.02 (0.05) | 0.16 (0.15) | 5 |
| | *p* | **<0.0001** | 0.08 | 0.72 | 0.70 | 0.30 | |
| **Model 2** | Estimate (SE) | 1.01 (0.17) | 0.31 (0.16) | 0.02 (0.08) | 0.02 (0.05) | – | 4 |
| | *p* | **<0.0001** | **0.04** | 0.82 | 0.61 | – | |
| **Model 3** | Estimate (SE) | 1.01 (0.17) | 0.33 (0.15) | – | 0.02 (0.05) | – | 2 |
| | *p* | **<0.0001** | **0.03** | – | 0.60 | – | |
| **Model 4** | Estimate (SE) | 1.08 (0.09) | 0.33 (0.15) | – | – | – | 0 |
| | *p* | **<0.0001** | **0.03** | – | – | – | |

as a male alternative mating tactic that is condition- and resource-dependent (*Albo & Costa, 2010*; *Albo et al., 2014*; *Trillo, Melo-González & Albo, 2014*). Our results suggest that following this, males probably reinforced the silk wrapping of worthless gifts, which in turn may act as a reliable signal of quality for females and allowed worthless gifts to successfully invade the populations (see *Albo et al., 2014*). Such an evolutionary scenario would imply paternity share not dependent on the gift content and hence, equal fitness success for males offering both gift types.

Sensory exploitation of female foraging motivation has been suggested as the origin of nuptial gifts in insects and spiders (*Sakaluk, 2000*; *Bilde et al., 2007*; *Albo et al., 2017*). Females accrue direct nutritional benefits from the gift and at the same time could use the gift as a signal of some aspect of male quality (*Zahavi, 1975*), such as hunting and provider abilities, which can also include more attractive sons (*Fisher, 1930*). Environmental limitations, however, seem to have led males to change gift content and produce worthless gifts diminishing the direct benefits for females (MJ Albo unpublished data), but maintaining the honest signal through silk wrapping (*Trillo, Melo-González & Albo, 2014*). This is not surprising as plasticity in sexual traits is predicted to be especially common in changing or ephemeral environments (*Cornwallis & Uller, 2010*). Riparian spiders, like *P. ornata*, generally feed on aquatic insects whose emergence is

**Table 4  Silk investment in worthless gifts when prey is present or absent.** Parameter estimates and *p*-values (with significance shown in bold) using Generalized Linear Mixed Models with a Gamma and Poisson distribution and a log link function to assess differences in the duration of silk wrapping and number of silk wrapping bouts for the worthless gifts produced by males in the Nutritive-Worthless group and the Worthless group. The initial model included also male size, trial number and the group*male size interaction. Model selection was performed by the Akaike Information Criterion (AIC).

| Silk wrapping duration | | Intercept | Group (Nutritive-Worthless) | Male size | Trial number | Group: male size | ΔAIC |
|---|---|---|---|---|---|---|---|
| Model 1 | Estimate (SE) | 1.38 (0.13) | 0.13 (0.19) | 0.15 (0.09) | 0.03 (0.03) | −0.04 (0.17) | 5 |
| | *p* | **<0.0001** | 0.48 | 0.08 | 0.42 | 0.82 | |
| Model 2 | Estimate (SE) | 1.39 (0.13) | 0.11 (0.17) | 0.14 (0.07) | 0.02 (0.03) | – | 3 |
| | *p* | **<0.0001** | 0.50 | 0.06 | 0.43 | – | |
| Model 3 | Estimate (SE) | 1.46 (0.08) | 0.11 (0.17) | 0.14 (0.07) | – | – | 1 |
| | *p* | **<0.0001** | 0.51 | 0.06 | – | – | |
| Model 4 | Estimate (SE) | −0.24 (0.71) | – | 0.42 (0.17) | – | – | 0 |
| | *p* | 0.73 | – | **0.01** | – | – | |
| **Number of silk wrapping bouts** | | **Intercept** | **Group (Nutritive-Worthless)** | **Male size** | **Trial number** | **Group: male size** | **ΔAIC** |
| Model 1 | Estimate (SE) | 0.53 (0.75) | −0.72 (1.92) | 0.19 (0.18) | 0.02 (0.04) | 0.16 (0.44) | 5 |
| | *p* | 0.48 | 0.71 | 0.30 | 0.60 | 0.71 | |
| Model 2 | Estimate (SE) | 0.41 (0.69) | −0.004 (0.15) | 0.22 (0.17) | 0.02 (0.04) | – | 4 |
| | *p* | 0.55 | 0.98 | 0.19 | 0.57 | – | |
| Model 3 | Estimate (SE) | 0.41 (0.63) | – | 0.22 (0.15) | 0.02 (0.04) | – | 2 |
| | *p* | 0.51 | – | 0.15 | 0.57 | – | |
| Model 4 | Estimate (SE) | 0.48 (0.62) | – | 0.22 (0.15) | – | – | 0 |
| | *p* | 0.44 | – | 0.15 | – | – | |

unpredictable and largely affected by floods (*Lytle, 2002*; *Iwata, Nakano & Murakami, 2003*; *Sanzone et al., 2003*; *Knight et al., 2005*; *Hagen & Sabo, 2014*). Thus, the fluctuating climatic conditions likely favour the evolution and maintenance of flexibility in the nuptial gift production optimizing males reproductive success (*Bussière et al., 2008*; *Kwan et al., 2008*; *Berger et al., 2014*; *Miller & Svensson, 2014*; *Montiglio et al., 2018*).

The trade-off between investing in prey capture or in silk wrapping is therefore a relevant assessment for males living under fluctuating environments. We did not find evidence in relation to the decision of males on gift type production as neither the size nor the courtship history (number of trials) appears to affect it. This result suggests that other factors, such as female attractiveness and/or condition, may play a role modulating the mating tactics (*Gaskett, 2007*). In fact, a recent study has found that *P. ornata* males can add more flies to the gift when females are in good body condition (*Solano-Brenes et al., 2021*). Though in our case all individuals were equally fed, we cannot rule out potential effects associated with variation in a female's condition. Nevertheless, our results regarding silk investment for worthless gifts give valuable information on this issue. We found that males spent more time and deposited more silk during worthless gift production than when the gift was nutritive, and that large males enhance the silk wrapping of the item as they are able to add more silk to the gift. Additional silk would be especially important

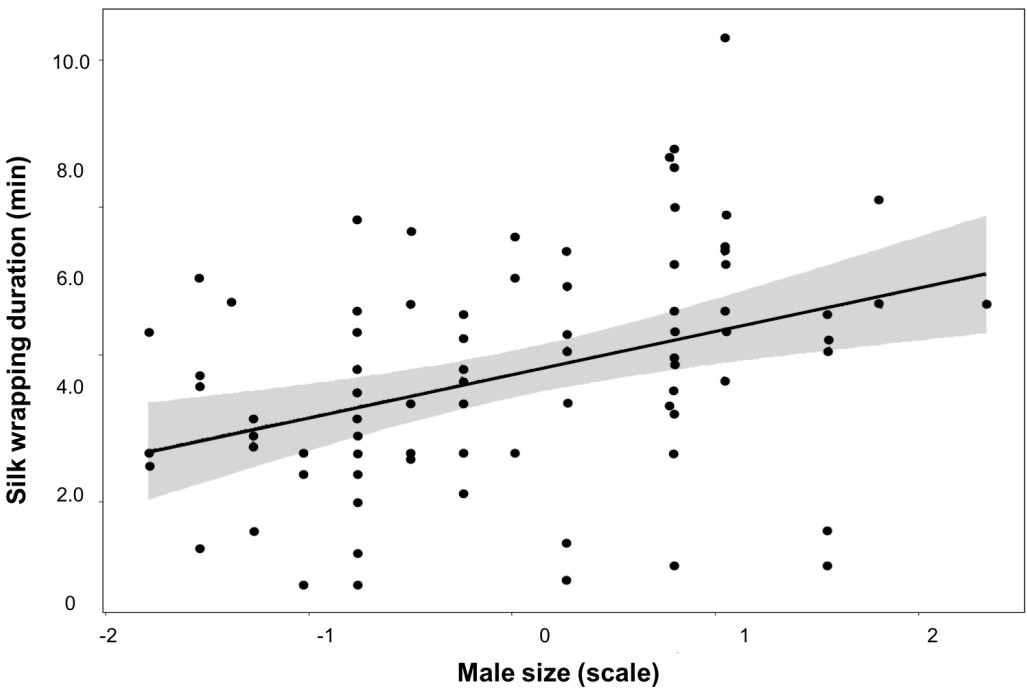

**Figure 3** **Silk investment in relation to male size.** Variation of silk wrapping duration according to male size for worthless gift production in *Paratrechalea ornata*. Data are shown as black points, linear predictor in black and the confidence interval in grey. These data are from males used in the two different experimental groups. Males from the Nutritive-Worthless group ($n = 18$) had the opportunity to produce nutritive or worthless gifts, and males from the Worthless group ($n = 18$) had the opportunity to produce only worthless gifts. Different letters indicate significant differences ($p < 0.05$).

when the gift is worthless, as can better camouflage its content and still attract females (visually and chemically) (*Brum, Costa-Schmidt & De Araújo, 2012*; *Trillo, Melo-González & Albo, 2014*). Silk may replace the edible content if it is nutritionally important but it is unknown whether females feed from the silk wrapping of the gift. On the other hand, when females receive worthless gifts they may try to leave the mating before the sperm transfer is completed. Hence, by adding more silk, males may acquire benefits in terms of controlling the mating duration by holding the gift with the claws of their third pair of legs during the sperm transfer thus reducing the likelihood the female will steal it, as is described for the closely related gift-giving spider *P. mirabilis* (*Andersen et al., 2008*; *Ghislandi et al., 2017*).

Overall, research indicates that silk wrapping is a relevant male trait in gift-giving spiders, especially when a male produces a worthless gift, but the cost of this strategy is unknown. *Ghislandi et al. (2017)* described worthless gifts as a cheap tactic (*Albo et al., 2011*; but see *Albo, Toft & Bilde, 2011*) in which males invest more silk to disguise the content, but are still able to increase their own body mass (despite this extra-silk investment) by eating the prey themselves. We doubt this is the same for *P. ornata* because, in the field, worthless gifts may be prey leftovers but very often include inedible items (like plant parts) (*Albo et al., 2014*). This means that males cannot feed from such non-prey items and thus they do not gain energy but rather probably spend energetic resources as further invest more in the

silk wrapping. Taken together these lines of evidence suggest that a male's silk investment is potentially an honest signal that females can use to measure male quality (*Albo, Toft & Bilde, 2011*; *Trillo, Melo-González & Albo, 2014*; *Macedo-Rego et al., 2016*).

This is a win-win scenario for males, but from the females' perspective, the worthless gift tactic may reduce their fitness success. Commonly, polyandry in gift-giving species is maintained throughout the food benefits gained by females from the gift during each mating (*Arnqvist & Nilsson, 2000*). We know that *P. ornata* females suffer from a reduction in fecundity if they accept multiple worthless rather than nutritive gifts (*Pandulli-Alonso, Quaglia & Albo, 2017*). Two scenarios are possible: either females are under exploitation suffering fitness costs and sexual conflict occurs (*Arnqvist & Rowe, 2005*) or their preferences for the gift content have co-evolved with male wrapping behaviours as, under natural conditions, they can supply food demands by increasing the foraging activity and thus their choice is likely cost neutral. It has been proposed that, in species with a high proportion of deceptive worthless gifts, female receptivity based on hunger state is unlikely (*Pandulli-Alonso, Quaglia & Albo, 2017*). There is no unequivocal study to resolve this puzzle. Some evidence suggests females may cryptically bias paternity towards males offering nutritive gifts (*Albo & Peretti, 2015*). Given this, offering a nutritive gift would provide fertilization benefits to males over offering a worthless gift and thus, selection would favour this tactic. This can explain the maintenance of nutritive gifts but not the high proportion of worthless gifts. Recent field studies have revealed that worthless gifts proportion not only vary among populations (38–96%) with different environmental conditions, but also within the mating season in a single population (MJ Albo unpublished data). This is the case of our study population in which males offering worthless gifts occur between 30–80% along the season. This variation in the mating tactic lead us to the discussion on whether offering nutritive gifts under certain dates might reward higher reproductive success for males than offering worthless gifts. How both gift-giving tactics are actually maintained is an open question.

## CONCLUSIONS

In summary, here we demonstrate that males are plastic in terms of whether they present a nutritive or a worthless gift. We suggest therefore that the gift-giving mating tactics were initially likely produced from a conditional strategy (*Gross, 1996*) related to prey availability and male condition. This phenotypic plasticity is revealed not only in the decision of using a prey or an inedible item and produce either a nutritive or a worthless nuptial gift, but also in the amount of silk deposited. This suggests that over evolutionary time the silk wrapping potentially become further relevant for female attraction, leading worthless gifts to invade the populations. The fact that the nuptial gift is composed by an exogenous (prey) and an endogenous (silk) part allows males to differentially allocate their energy and resources. There is a trade-off between investing in the exogenous part or in the endogenous one probably associated with fitness outcomes, which is relevant to understand whether the presentation of worthless and nutritive gifts result in similar fitness success in *P. ornata*.

## ACKNOWLEDGEMENTS

We thank Federico Reyes and Silvina Cortes for their help in the field and laboratory work. Laura Montes de Oca for her help in spiders' maintenance. We thank Sebastián Horta and Sistema Nacional de Areas Protegidas (SNAP) for the support to carry out this research in protected areas of Uruguay; Marina Castro INIA-La Estanzuela, Uruguay for housefly breeding material. We are grateful to Mauro Martínez, Bruno Buzatto, Laura Quintana and Daniel Naya for fruitful comments on the first draft.

### Funding

Maria J. Albo was supported by Caldeyro-Barcia National Science Award (MJA), PEDECIBA, and Postdoctoral fellowship IIBCE, Uruguay. Camila Pavón-Peláez was supported by Animal Behavior Society (USA). Valentina Franco-Trecu was supported by Postdoctoral fellowship PEDECIBA, Uruguay. Maria J. Albo and Valentina Franco-Trecu were funded by ANII (SNI). The funders had no role in study design, data collection and analysis, decision to publish, or preparation of the manuscript.

### Grant Disclosures

The following grant information was disclosed by the authors:
Caldeyro-Barcia National Science Award (MJA).
PEDECIBA, and Postdoctoral fellowship IIBCE, Uruguay.
Animal Behavior Society (USA).
Postdoctoral fellowship PEDECIBA, Uruguay.
ANII (SNI).

### Competing Interests

The authors declare there are no competing interests.

### Author Contributions

- Camila Pavón-Peláez performed the experiments, analyzed the data, prepared figures and/or tables, authored or reviewed drafts of the paper, and approved the final draft.
- Valentina Franco-Trecu analyzed the data, prepared figures and/or tables, and approved the final draft.
- Irene Pandulli-Alonso performed the experiments, prepared figures and/or tables, and approved the final draft.
- Therésa M. Jones conceived and designed the experiments, authored or reviewed drafts of the paper, and approved the final draft.
- María J. Albo conceived and designed the experiments, performed the experiments, prepared figures and/or tables, authored or reviewed drafts of the paper, and approved the final draft.

## Data Availability

The raw data is available at Mendeley: Pavón-Peláez, Camila; Franco-Trecu, Valentina; Pandulli-Alonso, Irene; Jones, Therésa M.; Albo, Maria J. (2021), "RawData Pavón-Peláez et al., PeerJ 2021", Mendeley Data, V2, doi: 10.17632/8cbmn7t84r.2.

## Supplemental Information

Supplemental information for this article can be found online at http://dx.doi.org/10.7717/peerj.12757#supplemental-information.

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
