# Peer review of "Beyond the prey: male spiders highly invest in silk when producing worthless gifts"

_PeerJ, doi:10.7717/peerj.12757_

## Round 0.1 · original submission · Major Revisions

You will see that both reviewers were broadly positive about the manuscript and this is also my impression. But at the same time, the reviewers raised a number of questions that should be addressed before I am able to reach a final decision. In particular, I would like to highlight the following points raised by them: the lack of clear reasoning that silk investment should be neutral, the lack of a clear explanation justifying why the conditions mentioned in the third paragraph of the introduction are necessary to favor the evolution of a high proportion of males providing worthless gifts, the possibility that prior experience may affect male behavior in subsequent exposures to females and some statistical approaches that should be revised.

Since both reviewers have made several constructive suggestions, I would like to invite you to respond to the reviewers' comments and send a revised version of your manuscript.

Best wishes,

Reviewer 1 ·

Basic reporting

Very good. Please double check grammar.

Experimental design

Excellent.

Validity of the findings

Good, but please see comments on the statistical analyses below.

Additional comments

This paper examines some of the costs and benefits of nutritive vs. worthless nuptial gifts in the trechaleid spider Paratrechalea ornata. The authors first ask whether males prefer to present worthless or nutritive gifts to females when both options are available, and whether this behaviour is repeatable. They also compare the investment in silk-wrapping of the two different kinds of gifts, and find that males invest more silk when preparing worthless gifts, whether or not there is a prey item available, and that silk wrapping is related to male size.

I really enjoyed reading this paper! It is very clearly written, the experimental design is very clever (especially the offering of the prey), and the results are interesting, adding to our understanding of the evolution of worthless gifts. The framing of the introduction and the discussion are excellent, and my main suggestions are to do with the data analysis and presentation.

Lines 104-108: I do not follow the reasoning that if the reproductive benefits of worthless and prey gifts are the same, the silk investment must be cost-neutral. I would expect that if the worthless gift has the same benefit as the prey gift, it must be because there is something added (like the silk investment) to cause the female to accept both kinds of gifts equally. I may be missing something, but please explain the reasoning here for others, who also may not follow.

The experimental design exposes males to the same group of females repeatedly. Could there be any effect of male or female experience on type of gift chosen (male) or female receptivity? Can you report whether males who switched tactics did so in the same direction (worthless first->nutritive later or vice versa) or switched back and forth?

I don’t think it’s appropriate to do model selection via AIC *and* then report p-values for specific predictors. My understanding is that one should do *either* model selection to come up with the best model that describes the relationship *or* report p-values from a model that includes the predictors one expects/predicts are influencing the dependent variable (decided a priori). Please see this blog post by Brian McGill for more discussion of this: https://dynamicecology.wordpress.com/2013/10/16/in-praise-of-exploratory-statistics/

I think that the data displayed in figure 1 could be more informative either by switching to a histogram-type graph with the number of males on the y-axis and percentage of nutritive gifts on the x-axis, or, rearranging the individuals in this graph by increasing proportion of grey, so that it is easier to see what proportion of males always gave worthless (or nutritive) gifts, and what proportion gave a mixture.

Please consider adding the actual data points to the box plots (e.g. points in grey behind see-through boxes) or violin plots instead of boxplots so that the reader can more easily see the distribution of the data. More suggestions for the best ways to visualize data here: https://journals.plos.org/plosbiology/article?id=10.1371/journal.pbio.1002128


Minor comments:

Please update figure captions with information about the name of the species and some details of the experimental context, etc. so that each display item can be understood in isolation from the rest of the paper. E.g., for figure 4, “Differences in silk wrapping duration according to body size for worthless gift production by Paratrechalea ornata males. Data are shown as black points, linear predictor in black and the confidence interval in grey. These data are from males (n = xx) used in two experiments, one where they had the option to make a nutritive or worthless gift, and another where they only had the opportunity to produce a worthless gift.”


Lines 20-21: please revise to “…offerING either a nutritive (prey) or worthless (prey leftovers) SILK-wrapped gift to females”

Line 55: “The divergence in an individual’s gift-giving plasticity can lead to the…” I have trouble understanding the meaning of this sentence, please clarify.

Reviewer 2 ·

Basic reporting

This is an interesting study that has potential to extend our knowledge on gift-giving behavior.
Manuscript structure conforms to PeerJ standards. The text is written in professional English language. Some corrections are needed (see below, in the Section 4. General comments, subsection 4.2 Other comments), but every reader is able to understand the study.
The authors provided all needed context, but the Introduction can be improved. The third paragraph of Introduction is too long (407 words), consequently it is not so easy to understand it. Authors have to reduce it. On the other hand, the fourth paragraph is too short (105 words). Given its content, I suggest adding the fourth paragraph to the fifth one (224). Combining the fourth and the fifth paragraphs will require some adjustments.
Methods are well presented, but some additional information should be provided (see below, in the Section 4. General comments, subsection 4.1 More relevant general comments). Data reporting requires more attention (see below, in the Section 3. VALIDITY OF THE FINDINGS).
The authors cite relevant literature. However, some statements lack references (L59-60; L73-75; L80-82).
Raw data is provided and it is easy to understand all information.
Figures are relevant and well described (but authors need to correct y-axis name in Figure 2B).

Experimental design

This is an original primary research within the scope of PeerJ. The research question is well defined, relevant and meaningful (e.g., do P. ornata males construct worthless gifts when given the opportunity to construct nutritive gifts?). The authors state how the research fills an identified knowledge gap (e.g., what are the implications of worthless gifts and what does it tell us about the evolution of gift construction?).
Authors allowed males to construct nutritive or worthless gifts along five trials. In order to run these trials, authors offered a living fly or exuviae to males and registered which option each male picked. This experimental design allows to: (a) test if males do construct worthless gifts when given the opportunity to construct a nutritive gift; (b) test if males are consistent in their behavior; and (c) compare the resulting nutritive and worthless gifts. In general, authors explain methods in detail (but a few adjustments should be made (see below in Section 4. General comments, subsection 4.1 More relevant general comments). Finally, the investigation seems to have been rigorous and ethical.

Validity of the findings

The authors found that: (a) males do construct worthless gifts when given the opportunity to construct a nutritive gift; (b) males are not consistent in their behavior; and (c) males invest more in terms of silk deposition when constructing worthless gifts.
All underlying data is provided. The reported results are particularly relevant once you acknowledge that the authors did not compare experimental groups according to the frequency of worthless gift construction or according to within-individual variance in gift construction. Appropriately, the authors did not make these comparisons because the experimental procedure is different for each group (in the Nutritive-Worthless group, males were offered a living fly via tweezers) and because the probability of worthless gifts occurrence is higher in the Worthless group from the very beginning. Therefore, the authors restricted the comparison solely to the worthless gifts constructed in each treatment group, circumventing the above-mentioned inappropriate comparisons.
In general, the data is robust and statistically sound. However, two results reported in the main text differ from the information provided in tables. First, Figure 2A and Table 3 show contrasting results. According to Figure 2A, wrapping duration was longer when males constructed worthless gifts. However, according to Table 3, wrapping duration was longer when constructing nutritive gifts (note the negative estimate for worthless gifts). Second, Figure 3 and Table 4 also show contrasting results. According to Figure 3, wrapping duration increased with male size. However, according to Table 4, wrapping duration decreased with male size (note the negative estimate for male size). Checking the data, it seems that the information in the main text is correct, and negative values were erroneously included in Tables 3 and 4. However, authors must revise the analyses that provided these results.
Aside from the comments above, authors need to improve data reporting. First, authors need to inform confidence intervals for estimates in Tables 1, 3, and 4. Second, while authors say that “Silk investment in worthless gifts did not differ between males in the presence (Nutritive-Worthless group) and absence of prey (Worthless group) (Table 4)” (L266-268, also see L270-272), Table 4 does not provide data that leads to this conclusion. To solve this problem, authors need to present estimates for each treatment group and trial number and need to present the AIC values for each model or for the top tested models. Third, a similar problem occurs in Table 3. Once again, authors need to provide AIC values for each model (or at least for the top models) and provide estimates for fixed effects (in this case, male size and trial number). Otherwise, statements made in the main text (L269-270) cannot be verified by the reader.
Finally, conclusions are linked to the original research question and limited to supporting results. Most conclusions are also well stated, but there are exceptions. The most important exception is found in the third paragraph of Discussion (L308-320; for other exceptions, see Section 4. General comments, subsection 4.1 More relevant general comments). In order to explain why males deposit more silk while constructing worthless gifts, the authors mention that silk (1) helps to attract females, (2) may be nutritious for females, and (3) enables males to better hold the gift. However, all these reasons apply for both nutritious and worthless gifts. In fact, avoiding the loss of the gift would be more important when holding a nutritious gift. Therefore, it is not clear why these three factors explain the higher investment in worthless gifts.

Additional comments

For clarity, I below provide suggestions directly to the authors.

4.1 More relevant general comments
(L38-39; L301-304) These sentences sound finalist/teleological. Why not try ‘In some invertebrate species, individuals (usually males) offer nuptial gifts to potential mates (usually females), increasing their reproductive success’ and ‘Thus, the fluctuating climatic conditions likely favour the evolution and maintenance of flexibility in the nuptial gift production, optimizing males reproductive success’?
(L55) What does “divergence in an individual’s gift-giving plasticity” stand for? This sentence could be clearer.
(L59) Here, you say that alternative tactics are less successful than the dominant strategy. However, this statement contradicts other statements in your text (e.g., L88-89; L94-97; L287-289).
(L107-108) Why does this rationale require that silk investment is cost neutral? You should clarify this part.
(L114-125) This paragraph should be improved in order to make predictions clearer. First, you present your hypothesis (L114). However, in the next sentences, you present scenarios that go against your hypothesis (L114-116; L116-118) or give support to your hypothesis (L120-123). It is interesting to show these contrasting scenarios, but it would be better to present them in a way that leads to your hypothesis.
(L118-120) You need to explain how this result can “indirectly demonstrate that males have no restrictions on their reproductive benefits in relation to the gift type”.
(L124) No reason is given for this comparison. Why is it interesting to contrast the silk investment in presence and absence of prey?
(L125) No reason is given for this prediction. Why do you expect no differences between presence and absence of prey?
(L184-185) Did you offer a live housefly to all males in the Nutritive-Worthless group? Or did one or more males capture an exuviae before you could offer the fly? In case you were able to offer a live housefly to all males, inform this in the text. Otherwise, explain why you included in your analyses males that captured an exuviae before having the opportunity to capture a fly.
(L192-194) It is not clear what ‘ending a silk wrapping’ stands for. How did you define the exact moment in which each male “ended the silk wrapping”? Please inform this in the text.
(L200-201) Please define a silk wrapping bout. Otherwise, one cannot completely follow your study protocol.
(L206-207) Because males in each treatment group differed in the type of gift they could produce, you did not exactly follow the same experimental protocol in both groups. I think you should inform that treatment groups differed in one single aspect (males in one group received an item via tweezers) and make implicit that you see no reason to expect that this difference affected male silk investment in worthless gifts. For example, you can say: Aside from offering a living fly via tweezers, “we followed the same experimental protocol described above, so males in both treatment groups had comparable courting opportunities but differed in the type of gift they could produce”.
(L218-220) It is not clear how the response variable was calculated. Is it the ratio between worthless gifts and nutritive gifts per male?
(L222-224) Male 2891 only constructed one gift. Is he included in this analysis? If yes, why?
(L308-310) A recent study tested if female quality influences male mating and reproductive effort (including investment in the wrapping procedure). See: Solano‑Brenes et al. Differential allocation in a gift‑giving spider: males adjust their reproductive investment in response to female condition BMC Ecol Evo (2021) 21:140, https://doi.org/10.1186/s12862-021-01870-1.
(L311-313) You say that you “found that males spent more time and deposited more silk during worthless gift production than when the gift was nutritive, and that large males can wrap the item better as they are able to add more silk to the gift”. Actually, you did not demonstrate that large males can better wrap gifts because they are able to add more silk to the gift. Therefore, this last part of the sentence “as they are able to add more silk to the gift” should be clearly presented as a hypothesis.
(L320-323) This sentence is disconnected to the previous one. However, the argument presented in the following sentences is convincing. Please, connect these sentences to the first part of the paragraph.

4.2 Other comments
In authors’ affiliations, some corrections are required. “Instittuto de investigaciones Biológicas Clemente Estable” should be “Instituto de Investigaciones Biológicas Clemente Estable”. Also, ‘montevideo’ lacks an initial capital letter.
Abstract is interesting and clear.
Throughout the text, sometimes you say trial, sometimes you say experiment. ‘Trial’ is more appropriate because you have one experiment and each male is used in five trials. Therefore, replace “experiment number” with “trial number”.
(L46) There should be no space between ‘e.’ and ‘g.’.
(L61; L66; L71; L72; L197; L267; L268) Please, include ‘i.e.,’.
(L69) It is not clear what ‘this’ stands for in the sentence “This rises because...”.
(L69) “for a female to consume complete the gift”. Please, revise this sentence.
(L79) First, it is likely worthless that
(L81) Early studies confirm that
(L90) quality, leading (include comma).
(L91) "preferences for the gift" is not clear enough.
(L104) To make clear that you are still talking about P. ornata, please say: “Based on the assumption that when a P. ornata male offers a worthless gift”.
(L105) Maybe replace ‘choose’ with ‘choosing’.
(L109) Consider saying ‘quality indicator’.
(L127) Materials & Methods
(L147) 21:00 and 23:00
(L150) Remove ‘were’.
(L154) field, with
(L166) co-occur, we
(L191) captured the fly
(L203) of prey, we
(L204) group, we
(L230-233) At first, R is in italic and in capital letter (L230), then, it is solely in lower case (L232-233). Please standardize it.
(L237) distributions
(L244) effects
(L251) ‘remaining’ is not appropriate because there are also the 19 trials in which males did not construct a gift.
(L259) (using nutritive or worthless gifts)
(L260) ‘also’? This is the only experimental group in which you tested repeatability.
(L261) bouts, differed
(L264-265) Consider saying: Comparing silk investment between both gift types by using the Nutritive-Worthless group showed that males invested...
(L266) No comma.
(L267-268) or in the absence of prey
(L271) from both groups, we found
(L275-276) Consider saying that ‘Here, we demonstrated that, even when P. ornata males had the opportunity to produce a nutritive gift, 20% of males chose to offer worthless gifts’.
(L287) Consider saying ‘(see Albo et al., 2014)’.
(L315) Replace ‘It’ with ‘Silk’.
(Figure 2B; Table 3; Table 4) Replace ‘number silk wrapping bouts’ with ‘number of silk wrapping bouts’.
(Figure 2) Figure 2A was uploaded in the online system as Figure 2, Figure 2B was uploaded as Figure 3, and Figure 3 as Figure 4.
(Table 2) Confidence intervals should be presented before the p value (i.e., in the second column).
(Table 4) produced by males in the Nutritive-Worthless group compared to the Worthless group.

---

## Round 0.2 · Minor Revisions

Dear authors,
You will see that both reviewers were highly positive about the manuscript, although they provided some small suggestions. For this reason, I would like to invite you to work on their comments and send a revised version of this manuscript.

Best wishes,

Reviewer 1 ·

Basic reporting

Good!

Experimental design

Very good!

Validity of the findings

Good.

Additional comments

I thank the authors for their careful consideration of my comments and those of the other reviewer. I think that they have addressed all of the comments adequately and I appreciate the updating of the figures. There are still a few places where the grammar could be cleaned up, a couple of examples below, but the writing is very good overall.

line 63: empidid should not be capitalized unless at the beginning of a sentence
line 107: change "within individual" to "within-individual"
line 133: should be "usually wraps..."

Also please note that the revised versions of the tables are in the tracked-changes version of the manuscript but not the pdf.

Reviewer 2 ·

Basic reporting

First, I congratulate the authors for this new version of the manuscript. As previously indicated, I think this study is a valuable contribution and I hope to see it published after a few adjustments. In the following, I present a few additional comments.
Manuscript structure and content in this new version conform to PeerJ standards. Once again, raw data is properly shared.
Introduction’s quality was undoubtedly improved. However, one sentence (L119-120) is disconnected from the previous ones. Thus, it is not clear how the inclusion of a second experimental group helps us to better understand male mating effort in this spider species. In my opinion, authors need to better connect L119-121 to the rest of the paragraph. Otherwise, Introduction is abruptly finished. OBS: small changes in the text suffice to solve this problem.

Experimental design

Methods are better described in this second version. Experiment design is appropriate.

Validity of the findings

As mentioned before, findings are meaningful.
Authors improved the quality of data reporting in several ways. However, it is crucial to correct a small (but highly important) mistake. In the Word document with tracked changes, authors correctly included the new versions of Tables 1, 2, 3, and 4. However, in order to build the pdf file containing all material, authors uploaded the old (and incorrect) versions of these four tables.
Authors also improved the quality of Discussion.

Additional comments

4.1 Most relevant general comments

(L56-62) I am not sure whether these statements on alternative mating tactics are accurate. In case they are accurate, maybe they should be clearer. For example, is it appropriate to say that “By definition, AMTs are a conditional strategy”? According to the same book chapter cited after this statement (i.e. Taborsky et al., 2008) “The relative importance of genetic monomorphism with conditional responses as opposed to genetic polymorphism for the evolution of ARTs has been extensively debated (Pienaar and Greeff 2003; see Gross 1996, Shuster and Wade 2003 for review). The vast majority of described cases of ARTs involves some conditional responses of reproductive competitors (Gross 1996, Lank et al. 1999). Because a conditional choice of tactics has been associated with genetic monomorphism, it has been argued that genetic polymorphisms play only a minor role in the causation of ARTs (Gross 1996, Gross and Repka 1998a). This view has been challenged (Shuster and Wade 2003)” (...) “Both a genetic polymorphism and a conditional expression of tactics with strong environmental influence during development have been found in different species of this group”. Additionally, is it appropriate to say that “By definition” (...) “the alternative tactic is less successful than the dominant strategy and therefore, typically occurs at a lower frequency within the population”? According to Taborsky et al. (2008): “In this case, genotype frequencies underlying ARTs are believed to be balanced by frequency-dependent selection, leading to equal lifetime fitness expectations of individuals using different tactics (Shuster and Wade 1991, Ryan et al. 1992, Repka and Gross 1995; but see Boxes 1.2 and 1.3 ; see also Chapter 2 of this book)”. I suggest that authors review this part of the manuscript (i.e., L56-62), which may lead to some corrections and/or clarifications, depending on their evaluation.

(L260-261) Values are not correct here. Instead of 55%, it should be 50%; instead of 6%, it should be 11% (as reported in Table 1).

4.2 Other general comments
(L51) It is written BussiÈre. Change it to Bussière.
(L195) Consider writing “(i.e., the last silk wrapping bout)”.
(L267) It should be “by using the Nutritive-Worthless group”.
(L278-279) This sentence starts in the past tense “even when P. ornata males had the opportunity to produce a nutritive gift” and continues in the present form “20% of them choose to offer worthless gifts”.
(L320) ‘its content’ instead of ‘it content’.
(L324) ‘receive’ instead of ‘received’.
(L328) It would probably be better to start a new paragraph here (in this current version, this paragraph is particularly long). The new paragraph would start with “Overall, research indicates”.
(Figure 1) Numbers in x-axis are not aligned with respective columns.
(Figure 2) Please, inform the meaning of ‘a’ and ‘b’.
(Figure 3) ‘Differences in silk wrapping duration according to Paratrechalea ornata male size for worthless gift production’ is not simple to understand. But more importantly, this figure does not exactly present differences in silk wrapping duration once there is no comparison between males and/or experimental groups. Instead, this figure shows how silk wrapping duration varies according to male size in Paratrechalea ornata.
(Table 1) Authors included confidence intervals for estimates in Tables 3 and 4, but not for estimates in Table 1. CIs or SE should be provided.
(Tables 1, 3, and 4) Authors forgot to substitute the term ‘Experiment’ for the term ‘Trial’.
(Tables 3 and 4) Sentences are duplicated.
(Tables 3 and 4) It would be better to report delta AIC values, setting the lowest AIC value to zero.
(Table 3; Silk wrapping duration; Model 1) Please, check the estimate value for gift type. Is it really a negative estimate?
(Table 4; Silk wrapping duration; Model 4) Please, include a space between -0.24 and (0.71).

---

## Round 0.3 · accepted · Accept

Dear Dr. Albo,

It is my pleasure to say that your manuscript has been accepted for publication in PeerJ!